# Production and Inhibition of Acrylamide during Coffee Processing: A Literature Review

**DOI:** 10.3390/molecules28083476

**Published:** 2023-04-14

**Authors:** Zelin Li, Chunyan Zhao, Changwei Cao

**Affiliations:** 1Department of Food Science and Engineering, College of Life Sciences, Southwest Forestry University, Kunming 650224, China; 2College of Food Science and Technology, Yunnan Agricultural University, Kunming 650201, China

**Keywords:** coffee, process stage, acrylamide, formation, inhibition strategy

## Abstract

Coffee is the third-largest beverage with wide-scale production. It is consumed by a large number of people worldwide. However, acrylamide (AA) is produced during coffee processing, which seriously affects its quality and safety. Coffee beans are rich in asparagine and carbohydrates, which are precursors of the Maillard reaction and AA. AA produced during coffee processing increases the risk of damage to the nervous system, immune system, and genetic makeup of humans. Here, we briefly introduce the formation and harmful effects of AA during coffee processing, with a focus on the research progress of technologies to control or reduce AA generation at different processing stages. Our study aims to provide different strategies for inhibiting AA formation during coffee processing and investigate related inhibition mechanisms.

## 1. Introduction

Coffee is the third-largest beverage worldwide and has been for >1000 years [1,2]. Further, it is the second largest commodity after oil in international trade, with more than 70 countries consuming coffee [1,3]. To obtain a high-quality cup of coffee, it has to go through some key processing procedures, including a selection of green beans, the drying process, roasting, grinding, and brewing [4,5]. After the coffee beans are dried, they are further roasted to obtain products of different degrees, such as light, medium, and deep, based on the consumer’s requirements [6]. Roasted coffee is further ground into powder, and different grinding and brewing techniques result in different sensory qualities [7,8,9]. However, these key processes can produce acrylamide (AA), which directly affects the quality and safety of the coffee [10].

AA is a common compound in high-temperature processed food that has genetic, toxic, and carcinogenic properties and can cause nerve damage [11]. AA is mainly produced by the reaction of carbonyl compounds with amines during high-temperature processing [12]. It is considered a Group 2A probable carcinogen by the International Agency for Research on Cancer, with excessive intake having a negative effect on human health [13]. AA or its precursors can be formed at each coffee processing stage because carbonyl and amino compounds are present, get accumulated during processing, or coffee oil is degraded during storage and transportation [14,15]. For example, in the drying stage, a small amount of AA precursor was produced under the action of asparaginase; these precursors are positively associated with AA content [16]. The proteins, amino acids, fats, sugars, and other substances contained in coffee beans can also result in the formation of some toxic components, including AA and 5-hydroxymethylfurfural (HMF) [17,18].

Owing to the negative effects of AA, several countries and regions have limited AA content in coffee products. For example, the European Regulation 2017/2158 recommended a maximum AA content of 400 μg/kg in roasted coffee, 850 μg/kg in instant coffee, and 500 μg/kg and 4000 μg/kg in coffee substitutes based respectively on cereal and chicory [10]. Therefore, it is necessary to reduce or inhibit AA formation during coffee processing. However, it has always been a challenge to control AA content in food without affecting its final quality, especially in quality control and AA suppression in coffee production. Taken together, the present paper aims to review the formation and inhibition strategies for AA in different coffee processing stages to provide some useful information regarding AA control in coffee processing and production.

## 2. Formation, Hazards, and Determination of AA Content in Coffee

### 2.1. Pathway of AA Formation in Coffee

Temperature is the main condition for AA production. Once the temperature exceeds 120 °C, AA is generated. Coffee roasting is performed at a temperature of >150 °C, thereby creating a favorable precondition for AA formation [19,20,21]. The three main AA formation pathways in coffee are presented in Figure 1.

First, lipid reactions are an important indicator affecting the quality of coffee and could lead to the production of AA (Figure 1a). Improper treatment during roasting, storage, processing, transportation, and other stages can result in the oxidation and corruption of coffee lipids, further leading to the production of harmful compounds and affecting cholesterol conversion [22,23]. AA can be formed by a dehydroxylation reaction between acrolein and acrylic acid generated after the decomposition of lipids and existing amino acids or ammonia formed during the pyrolysis of proteins [24,25]. Furthermore, acrylic acid can generate AA by reacting with ammonia. Fortunately, the formation of AA via acrolein and acrylic acid is limited because of the low release and capture of free ammonia in coffee and the high processing temperature.

Second, the heat reaction of protein and amino acids can also generate AA (Figure 1b). The proteins in coffee, particularly aspartic acid, were degraded to free amino acids. Then, aspartic acid underwent decarboxylation and deamination to form AA under high-temperature conditions [1,26]. Given that extremely high temperatures are applied during coffee roasting, the amino acids also participate in the formation of melanoidins, which contribute to the black color of coffee.

The Maillard reaction is the most common pathway to produce AA in coffee via free amino acids and reducing sugars (Figure 1c). Asparagine and sucrose (0.30–90 mg/g) in coffee are essential precursors involved in AA formation. Although sucrose is not a reducing sugar, its decomposition in the early stage of coffee roasting can directly affect the synthesis of AA by generating new compounds containing carbonyl groups or decomposing into low-molecular-weight reducing sugars [27,28,29]. The unstable Schiff base, which follows the thermal reactions of dehydration and N-glycation conjugation condensation of asparagine and reducing sugars, rapidly reorganizes to form 3-amino propanamide (3-APA), followed by a β-elimination reaction of decarboxylation to produce AA [30]. Sucrose degradation under low moisture conditions occurs via glycosidic bond cleavage, resulting in the formation of glucose and fructofuranosyl cations [31]. Dehydration of glucose results in the formation of α-dicarbonyl compounds, which may react with asparagine to produce AA [32]. Once heated, the sugar also forms a furan compound in coffee, which is also another precursor for AA formation [33,34,35]. In other words, under high-temperature conditions, the sugar is decarboxylated via the oxazolidine-5-ketone pathway to form HMF; further deamination, condensation, and hydrolysis generate AA [36,37,38]. HMF is also formed during the caramelization process via the elimination of three molecules of water from glucose [32].

### 2.2. Potential Hazards of AA

The potential toxicity of AA to organisms has been widely reported and verified. After long-term low-dose intake of AA, symptoms, including skin peeling and erythema, numbness of limbs, hyporeflexia, and peripheral neuropathy, can appear. Further, long-term ingestion in animals can result in ataxia, muscle weakness, and dyskinesia [39,40,41]. The potential toxicity of AA to the body and its mechanism are presented in Figure 2.

a.Neurotoxicity: The central nervous system is an important site of active oxygen metabolism in the body. As described in Figure 2a, long-term intake of AA can induce reactive oxygen species (ROS) to constantly attack cell membrane lipids, proteins, and DNA, damage the main target organs, and induce diseases such as Alzheimer’s and Parkinson’s disease [42]. Some studies have reported that AA induced an increase in the levels of oxidative stress-related enzymes such as superoxide dismutase (SOD), glutathione peroxidase (GSH-Px), and catalase (CAT) in the peripheral blood and brain [43]. Furthermore, it induced the destruction of the structure or function of the peripheral nervous system, resulting in the weakening or disappearance of movement and sensation [44].b.Immunotoxicity: AA can also stimulate the immune system to produce immune responses and activate mitogen-activated protein kinase (MAPK), nuclear factor-κB (NF-κB), and other related pathways for defense, as shown in Figure 2b. Some studies have reported that treating human neuroblastoma cells with AA could activate the extracellular signal-regulated protein kinase (ERK) to induce the death signaling pathway, c-Junn terminal protein kinase (JNK), and p38 mitogen-activated protein kinase (p38 MAPK) pathways, and upregulate the expression of proapoptotic proteins, resulting in cell apoptosis [45,46].c.Reproductive toxicity: AA has also been proven to exhibit reproductive toxicity (Figure 2c). After being catalyzed by the cytochrome P450 enzyme, AA is epoxidized to form glycidamide (GA). Then, AA and GA react with protamine in the testis to produce S-(2-formamido-2-hydroxyethyl) cysteine and S-carboxyethyl cysteine, eventually affecting fertility [47,48]. AA can damage the reproductive system by damaging normal Sertoli cells in male rats and the function of Leydig cells, as well as induce the abnormal secretion of testosterone and luteinizing hormone, resulting in abnormal sperm-related gene expression, decreasing the number of sperm to reduce the activity of sperm, and increasing the sperm deformity rate [49,50]. Further, AA can induce ovarian dysfunction in female Wistar rats by upregulating apoptosis-related genes [51].d.Other toxicities: AA can damage the liver, kidneys, lungs, bladder, and digestive tract and may even cause testicular mesothelioma, adrenal cortical adenoma, astrocytoma, and oral tumors [52]. At present, there are no studies on the harmful effects of AA in coffee on the human body; however, its toxic effects in food have long been confirmed via animal experiments or in vitro experiments using human cells. A AA toxicity test in rats in early 2005 revealed an LD_50_ of 107–203 mg/kg·bw and confirmed that AA has low toxicity [53]. Nevertheless, some studies have reported that the harmful effects of AA on the human body were mainly reflected as damage to human immune function, nervous system, genetic material, mitochondrial dysfunction, mutation, genotoxicity, as well as its potential carcinogenicity [54,55]. For example, the majority of GA-induced mutations in human tumors occurred at the A:T base pairs, with AT > TA and AT > GC mutations on specific *TP53* codons [56]. In other words, DNA adducts provide a possible mechanistic basis for mutation types and mutational signatures occurring following GA treatment, a reactive metabolite of AA [57]. Thus, the European Regulation has advised a maximum AA content of 400 µg/kg in roasted coffee.

### 2.3. AA Detection

Considering the toxicity of AA, it is crucial to precisely evaluate and reduce its concentration in coffee to mitigate potential risks to consumer safety. Nevertheless, direct quantitative analysis proves challenging owing to the absence of prominent chromophoric groups, such as conjugated double bonds in aromatic rings and conjugated triple bonds [58,59]. Consequently, sample pre-treatment is typically required. Presently, methods for the determination of AA in coffee can be categorized as traditional and novel. Traditional detection techniques encompass gas chromatography (GC), gas chromatography-mass spectrometry (GC-MS), high-performance liquid chromatography-mass spectrometry (HPLC-MS), high-performance liquid chromatography-mass spectrometry/mass spectrometry (HPLC-MS/MS), and capillary electrophoresis (CE) [39]. Novel detection techniques encompass enzyme-linked immunosorbent assay (ELISA), biosensors (electrochemical biosensors, fluorescence sensors), spectroscopy, highly sensitive gas chromatography detectors (high-resolution time-of-flight mass spectrometry, nitrogen-phosphorus detectors, tandem mass spectrometry), inhibition-based spectrophotometry, and machine vision approaches [60,61]. While there are limited studies employing these novel detection techniques for AA analysis in coffee, they have been extensively utilized for food detection purposes.

Traditional methods offer rapid, sensitive, and accurate analysis, making them suitable for the determination of ultra-trace AA in food. However, sample derivatization is necessary to enhance the stability. In a study by Ku Madihah et al. [62], the researchers employed a solid-phase extraction C18 column, conditioned with 3 mL of acetone and 3 mL of formic acid, to extract AA from coffee powder. The analysis was conducted using gas chromatography equipped with a Flame Ionization Detector at a temperature of 260 °C. The result indicated an AA concentration of 0.23 mg/100 g. Novel detection techniques not only preserve the rapid, sensitive, and accurate characteristics of traditional detection but also address the uncertainty stemming from the required derivatization to some extent. By employing a 3-mercaptobenzoic acid-AA-bovine serum protein (BSA) complex to generate antibodies and utilizing an indirect competition approach for AA detection in food via ELISA, recovery rates of 95–100% can be achieved [63]. A rapid and dependable technique for assessing AA content in fried foods employs surface-enhanced Raman spectroscopy. This method demonstrates a detection range of 5–100 μg/kg, with limits of detection and quantification at 2 μg/kg. Acrylamide recoveries range from 73.4% to 92.8%, showing consistency with the conventional LC-MS/MS method [64]. Table 1 lists some methods for detecting AA in coffee.

## 3. Control AA Production in Processing Stages

AA production can potentially occur at every stage of coffee processing; therefore, it should be stringently controlled using appropriate techniques at each stage. Figure 3 summarizes the methods of inhibiting AA formation during various stages, such as variety selection, drying, processing, roasting, storage, and brewing, as reported in several studies.

### 3.1. Variety Selection

AA formation in food during processing necessitates the involvement of one or more precursor substances. Consequently, selecting raw materials with few precursors is an effective approach to reducing the final acrylamide content, particularly in products such as coffee, which rely on a single raw material for production and processing (Figure 3a) [10]. Arabica, Robusta, and Ribirica are the three primary coffee varieties globally. Arabica and Robusta beans are commonly chosen as commercial coffee beans in the processing industry, representing approximately 64% and 35% of the world’s coffee production, respectively. Ribirica coffee, however, is not utilized as a commercial bean due to its distinct flavor and size [5,74]. Furthermore, these coffee varieties exhibit differences in their composition, including volatile substances, chlorogenic acids, diterpenoids, caffeine, and other constituents, as well as distinct sensory properties [75]. As depicted in Figure 3a, coffee varieties also differ in carbohydrate content, apart from the aforementioned substances. Arabica coffee beans contain higher levels of sucrose (74.4 ± 8.5 mg/g) and lower concentrations of asparagine (486 ± 97 μg/g) and alanine compared to Robusta coffee. Consequently, Robusta coffee can supply a greater number of precursors for the ultimate formation of AA [76,77]. Regarding AA precursors, agronomic factors such as cultivation practices, soil type, fertilization, pest control, irrigation, and climate conditions significantly influence their composition in green coffee. In high-temperature environments, mature Arabica coffee beans contain higher asparagine levels compared to those in lower-temperature settings [78].

Table 1 illustrates the differences in AA content among coffee varieties. Robusta coffee generates approximately 1.7–2.9 μg/30 mL of AA, which is twice the amount produced in Arabica coffee samples [20]. When Robusta and Arabica coffee beans are roasted at 250 °C for 7.5 min, the AA content in the former exceeds 3500 ng/g, while the latter contains less than 500 ng/g [67]. Similarly, Lantz et al. [79] observed that the AA production in Robusta coffee beans during processing was, on average, 34% higher than in Arabica coffee beans. Interestingly, despite the low asparagine content in Arabica coffee beans, the AA levels were high following the roasting [21]. This outcome may be attributed to the thermal decomposition of sucrose in Arabica coffee, resulting in a substantial increase in reduced sugar content. It was found that the content of reduced sugar in roasted Arabica coffee beans was five times higher than that in green coffee beans. This observation confirmed that AA production levels are indeed associated with the content of precursor substances in coffee beans. Therefore, selecting coffee bean varieties with lower precursor content is a viable strategy to reduce AA intake at the source.

### 3.2. Drying Process Stage

The postharvest drying of fresh coffee cherries not only plays a crucial role in delivering a stable product and releasing coffee flavor but also influences the concentration of AA precursor [5]. Several studies have demonstrated that wet-processed Arabica coffee beans exhibit lower concentrations of glucose, fructose, and other sugars compared to their dry-processed counterparts. However, sucrose levels remain unaffected by the processing methods [7]. This finding suggests that dry-processed coffee contains a higher number of precursors, increasing the likelihood of acrylamide formation in the final product [80]. Another factor contributing to the relatively elevated AA content in coffee is the blending of immature and defective beans during the drying process. Both immature and defective beans possess higher concentrations of free asparagine [81]. Asparagine is the primary amino acid in immature coffee cherries. Its concentration in wet-process coffee is 0.03 g/100 g lower than that in dry-process coffee. This difference arises because the wet process accelerates the metabolism of asparagine in coffee beans [81].

Free asparagine in coffee serves as a limiting factor in the ultimate formation of AA. As illustrated in Figure 3b, pretreatment of coffee beans in the drying stage can reduce the content of AA precursors. Moreover, the hydrolysis of free asparagine into aspartic acid and ammonia, catalyzed by asparaginase, effectively reduces the concentration of free asparagine in green coffee beans [25]. Treatment with exogenous asparaginase produced by *Aspergillus oryzae*, at varying concentrations, can effectively decrease the levels of asparagine and AA. Specifically, asparagine content can be reduced by 70–80% and acrylamide content by 55–74% [82]. In a study conducted by Correa et al. [1], the use of asparaginase to treat Arabica coffee beans resulted in a significant decrease in AA content. The concentration of asparaginase used ranged from 1000 to 5000 ASNU/kg, and the reduction in AA content was between 77% and 85%. The treatment had no effect on major substances such as caffeine, chlorogenic acid, and caffeic acid. Another study also reported a decrease in asparagine content by 30% when 3000 ASNU of asparaginase was added to Arabica and Robusta coffee beans [83]. Several patents have reported on the combination of asparaginase with decompression and pressure technology to facilitate the enzyme’s penetration into the dense and impermeable green coffee beans, ultimately inhibiting further AA production [84]. The potential effect of these treatments on the sensory properties of the final processed coffee beans has not been thoroughly evaluated. Additionally, enzymatic treatment requires additional processes involving steam or wetting, which can lead to the need for re-dehydration of the coffee and an increase in operational procedures and costs.

In recent years, decaffeination of green coffee beans has emerged as an additional stage to meet special needs. This stage involves pre-wetting the dried coffee beans with water, followed by the removal of caffeine using solvent extraction, water extraction, or pressurized carbon dioxide, and subsequent drying of the beans in the industrial decaffeination process [28]. It is possible that the decaffeination process does not affect the AA precursors, and therefore, the AA content may not change significantly in the final roasted coffee [85].

### 3.3. Roasting Stage

Roasting is a crucial step in the processing of coffee that enables the release of coffee aroma compounds and is essential in determining the quality characteristics of the coffee [86]. Coffee roasting is a process that involves a high-temperature treatment, typically above 200 °C, and the duration and temperature of the roasting process depend on the desired sensory quality, aroma, and color of the final product [87]. In addition, the extent of roasting, whether it is light, medium, or deep, plays a crucial role in determining the AA content in the coffee and can impact the sensory properties of the final product. Moreover, the duration of the roasting process also influences AA formation [14]. Thus, regulating the temperature and time of roasting, employing exogenous additives, and utilizing diverse roasting techniques can help to mitigate the content of AA to some extent (Figure 3c).

The AA formation in Robusta and Arabica coffee beans during roasting was greatly affected by different temperatures (220–260 °C) and times (5–15 min). At the beginning of roasting for 5 min, the maximum AA content in the 2 of varieties coffee beans was 3800 ng/g and 500 ng/g, respectively. However, with an increase in roasting temperature and time, the level of AA began to decrease. At 260 °C and 15 min, the level of AA decreased to 708 ng/g and 374 ng/g, respectively [76]. The AA formation in coffee during roasting is a dynamic process. For instance, when roasted at 200 °C for varying times (5, 10, 15, 30, and 60 min), the AA content initially reached a peak value of 468 μg/kg after 5 min and then gradually decreased until it was undetectable after 60 min [65]. Interestingly, the levels of HMF, 2,4-heptadienal, 4-hydroxynonenal, and 4,5-epoxy-2-decenal in roasted coffee beans were found to increase with the duration of roasting. Notably, the level of HMF was observed to reach as high as 485 μg/g at 10 min [65]. The AA content in lightly roasted coffee was higher but decreased when roasted over 4 minutes at 236 °C [88]. At temperatures above 200 °C, the time required to reach the peak AA content decreased with increasing temperature. On the other hand, at temperatures below 150 °C, the AA content increased with longer roasting times [68]. A recent study has confirmed that the formation of AA in both Arabica and Robusta coffee varieties was mainly observed in the initial 10 min of the roasting process, followed by a gradual rise in temperature and duration [89].

The formation of AA during roasting is known to have negative effects on both coffee quality and human wellness. Therefore, it is necessary to develop strategies to reduce or inhibit AA production. One effective approach to obtaining high-quality coffee with low levels of AA is to use a central combination design to optimize the roasting conditions of Robusta coffee samples. By doing so, it was possible to achieve a reduction in AA content of up to 0.23 mg/100 g after optimization [62]. Based on most studies, the AA content in coffee generally decreased with the increase in roasting time and temperature, especially at temperatures above 200 °C. Therefore, high temperatures and sufficient roasting time are necessary conditions for reducing the AA content. In addition, superheated steam roasting can also effectively decrease AA content, with reductions of 63%, 29%, and 25% observed at roasting temperatures of 210 °C, 230 °C, and 250 °C, respectively, for dark roasts. However, the specific AA content may vary due to different process parameters and coffee varieties [90]. Supercritical CO_2_ extraction is a modern and environmentally friendly method for extracting organic compounds from solid food matrices. Banchero et al. [91] employed this technique to eliminate AA from roasted coffee, and the findings revealed that this method did not alter the caffeine content of coffee. The maximum extraction efficiency of AA was observed to be 79%.

In addition to optimizing roasting time and temperature, the physicochemical properties of AA can also be utilized to reduce its content. AA has a boiling point of 125 °C at 3.33 kPa, which allows for its volatilization under vacuum conditions. A study has demonstrated that using a static oven with a vacuum pump for low-pressure treatment and roasting of coffee samples at 200 °C can reduce the AA level in coffee by 50% compared to traditional roasting [34]. The physicochemical properties of AA, such as its low molecular weight and high polarity, make it easily extractable using a non-toxic organic solvent. A combination of a 9.5% ethanol solution and supercritical fluid extraction at 100 °C at 200 Pa for 1035 min has been reported to result in an extraction rate of 79% [91]. However, these methods for reducing AA content during roasting have rarely been evaluated for their impact on sensory properties, and the mechanisms of AA degradation and the possibility of producing other harmful substances have not been clearly established. Table 2 summarizes additional literature on methods for reducing AA content in coffee during roasting.

### 3.4. Storage Stage

Similar to other heat-processed foods, the AA content in coffee is not stable during storage and packaging. The decomposition of AA is affected by environmental conditions, such as time, temperature, and air composition, during storage and packaging. The storage temperature is an important factor affecting the variation of AA. The AA degradation during storage is related to the presence of -SH radicals in coffee. This is because -SH can undergo a Michael addition reaction with the vinyl group of AA [92]. Figure 3d illustrates that the degradation of AA in coffee is affected by storage time and temperature. For instance, in the sealed storage of instant coffee and coffee substitute at 25 °C, the AA content decreased from 28% to 33% after 12 months. However, when stored at 4 °C, the AA content barely changed after 6 months and 12 months due to the inhibition of AA degradation under low temperature and high -SH free radical conditions [93]. The AA content can also decrease by 40–65% in coffee samples after opening the package and storing it at room temperature for 6 months, but no change in AA content was observed after storing it at −40 °C for 8 months [94]. In another study, coffee samples stored at different temperatures (–18 °C, −4 °C, room temperature, and 37 °C) for 12 months showed that the lowest AA content was observed at 37 °C [95]. Moreover, AA may react with the nucleophilic amino groups of amino acids from the proteinaceous backbone of melanoidins via the Michael addition during long-term storage [96].

The content of AA in coffee is also affected by storage time. Previous studies have confirmed that the AA reduction during long-term storage can be described by the second-order reaction kinetics equation, with an activation energy of approximately 73 kJ/mol [79]. Baum et al. [97] observed that AA content decreased significantly after 16 weeks of storage at 37 °C, and this degradation was found to be associated with the covalent binding of insoluble substrates. Radioisotope ^14^C was used as a tracer to track the change of AA during coffee processing. Although some hypotheses exist regarding the degradation mechanism of AA during storage, they have not been fully elucidated. Furthermore, the -SH radical is an important taste radical in coffee. Whether its reduction after storage affects the flavor substances or sensory quality of coffee requires further investigation.

### 3.5. Brewing Stage

AA is a water-soluble compound that readily dissolves into the coffee during brewing [94]. Preventing the extraction of AA during brewing is crucial to avoiding its exposure during coffee consumption. Various factors, such as the fineness of coffee grinding, the ratio of powder to water, pressure, extraction temperature, and time, can greatly affect the dissolved level of AA during brewing, depending on regional culture, consumption habits, brewing methods, and technologies [98]. The content of AA in instant coffee was found to be higher (ranging from 16.5 to 79.5 ng/mL) than that in other coffee products, such as Turkish ready-to-drink coffee, although the difference was not statistically significant [99]. A study on the effect of brewing time on AA content in coffee showed that the lowest AA concentration (4.0 ng/mL) was obtained after brewing with cold water for 3 h [100].

Apart from controlling brewing time and other conditions, the use of unconventional treatment methods could also help reduce the level of AA. This is because AA degradation is linked to brewing temperature and coffee concentration, and the acrylamide enzyme can hydrolyze AA into acrylic acid and ammonia. For instance, the direct addition of varying amounts of the acrylamide enzyme to instant coffee during infusion and extraction at both 37 °C and 70 °C for 30 min has been shown to reduce the AA levels, but the impact on sensory and safety quality needs further investigation [101]. A study showed that treating instant coffee with an immobilized acrylamide enzyme for 60 min could also result in the degradation of AA, which was consistent with the batch-based degradation kinetics of AA [102]. Moreover, the addition of yeast or various additives could also reduce the AA content in coffee. This approach can be implemented in the production and processing of regular canned coffee, and it holds great potential for further development (Figure 3e) [14,70,103]. Further verification is needed to evaluate the impact of these coffee treatments on sensory characteristics and safety.

## 4. Conclusions and Prospect

Controlling the AA content in coffee is crucial for ensuring its quality and safety. To achieve this, each stage of coffee processing should be carefully managed. Starting with the selection of green beans, arabica coffee can be chosen as an excellent option. Moreover, the wet processing method can be employed to generate lower AA precursor content and to degrade precursors using proteases. Roasting is the main stage where AA is generated, and controlling the roasting temperature (220 °C) and time (over 10 min) can significantly reduce the AA content. Additionally, applying pressure during coffee roasting can also help lower the AA content. Storage time and brewing temperature also affect the AA content in coffee beverages. Long storage times (12 months) at 37 °C and brewing at a lower temperature (4 °C) for a relatively longer duration (such as 3 h) can reduce the amount of AA dissolved in coffee beverages. However, while previous studies have mainly focused on reducing AA production, it is necessary to investigate other toxic substances, evaluate coffee quality, explore inhibition mechanisms, and develop the practical application of new technologies to produce high-quality coffee with lower AA levels in the future.

## Figures and Tables

**Figure 1 molecules-28-03476-f001:**
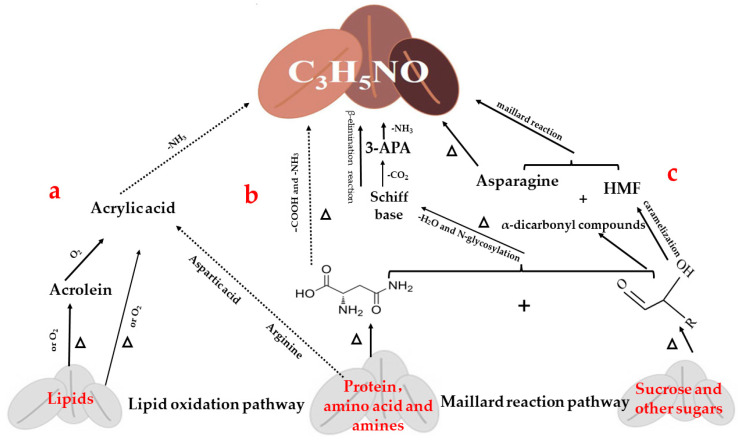
Potential pathways of acrylamide formation during coffee processing. (a) Lipid oxidation pathways; (b) Protein and amino acid degradation pathways; (c) Maillard reaction pathway.

**Figure 2 molecules-28-03476-f002:**
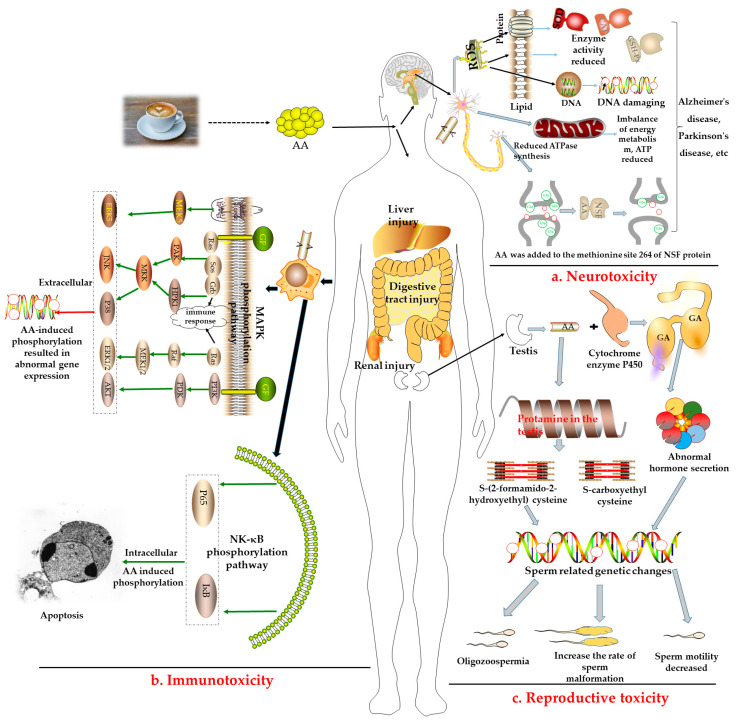
Mechanisms of AA-induced toxicity.

**Figure 3 molecules-28-03476-f003:**
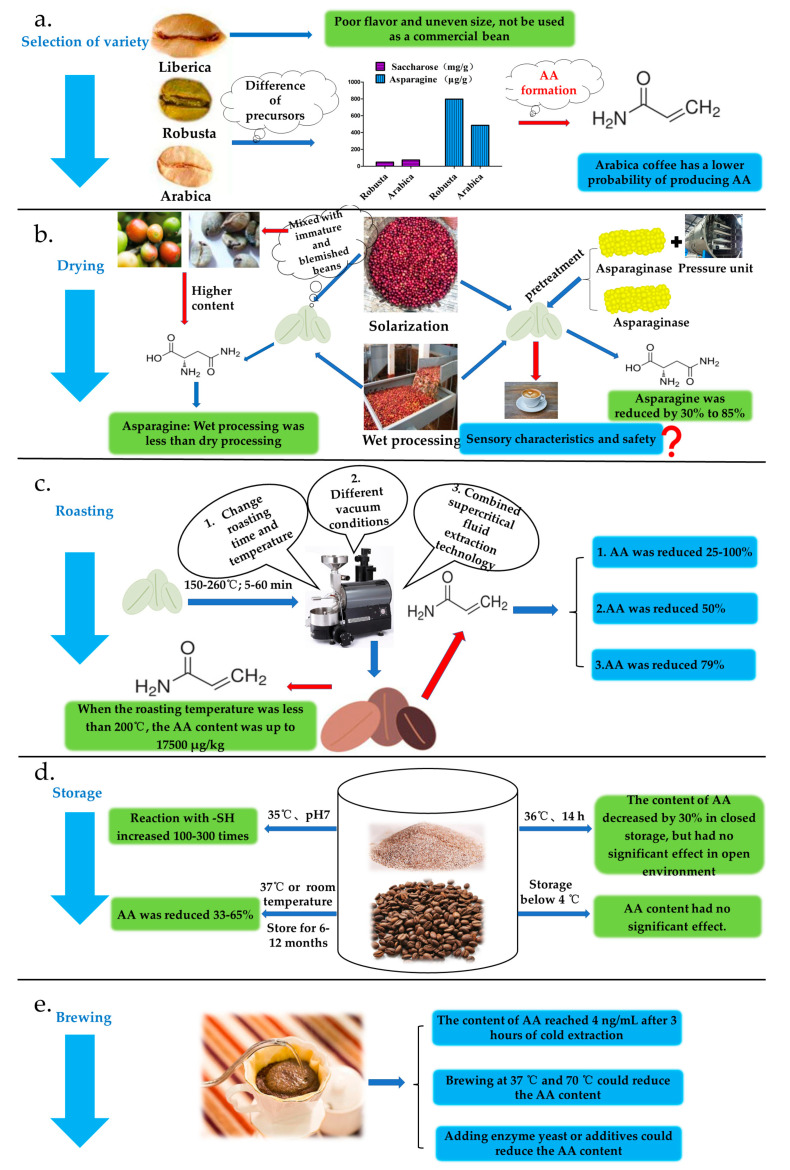
A review of studies investigating acrylamide inhibition during coffee processing. (**a**) Selection of variety; (**b**) Drying process stage; (**c**) Roasting stage; (**d**) Storage stage; (**e**) Brewing stage.

**Table 1 molecules-28-03476-t001:** A summary of studies in the literature regarding acrylamide detection methods in coffee.

No.	Coffee Samples	Methods	Treatment	Content (μg/kg)	Reference
1	Arabica, Robusta	GC-MS	210 °C roasted 8–11 min	0.87–2.92 μg/(30 mL espresso)	[20]
2	Arabica, Robusta	LC-MS/MS	Wet drying (5 degrees of roasting)	400–1130	[21]
3	Robusta	GC-FID	Wet drying (180–202 °C roasted)	1000–17,500	[62]
4	Arabica	LC-MS/MS	220 °C roasted	468	[65]
5	Instant coffee	Inhibitory reduction spectrophotometry	Hot water dissolved	888.3	[66]
6	Arabica, Robusta	LC-MS	Wet drying (220–260 °C roasted)	90–500	[67]
7	Arabica	LC-APCI-MS	150, 200, and 225 °C roasted	50–500	[68]
8	Arabica, Robusta	LC-MS/MS	Six ways to roast	130–480	[69]
9	Arabica	LC-MS/MS	Medium roast	1020	[70]
10	Instant coffee	LC-HRMS	Dissolve ultrapure water mixed with AA internal standard	159	[71]
11	Arabica	Stable isotope dilution and LC-MS/MS	Roasted	22.2–326.4	[72]
12	Robusta and Arabica	GC-MS	200–245 °C	159–484	[73]
13	Instant coffee	LC-ESI-MS	Dissolve ultrapure water mixed with AA internal standard	41–1049	[74]

**Table 2 molecules-28-03476-t002:** Study on the inhibitory effect and strategy of AA in coffee during roasting.

No.	Samples	Roasting Conditions (T: °C, t: min)	Treatment	Optimal Inhibition Conditions	Inhibition Ratio	Reference
1	Arabica	T: 5, 10, 15, 30 and 60	0.15 kPa	200 °C roasted 10 min	50%	[34]
2	Robusta	T: 200, t: 10 tradition, tradition-vacuum combined and vacuum	Different roasted temperatures and times	210 °C roasted 40 min	90.92%	[62]
3	Arabica	T: 220	Different roasted times	220 °C roasted 60 min	100%	[65]
4	Turkey Arabica	T: 150–210	Different roasted temperatures and times	225 °C roasted 30 min	43.48%	[68]
5	Robusta, Arabica	T: 15–40	Different roasted temperatures and times	260 °C roasted 15 min	81.37% and 25.2%	[76]
6	Robusta, Arabica	T: 150, 200, 225	Different roasted temperatures and times	236 °C roasted 10 min	42.86% and 57.14%	[88]
7	Robusta, Arabica	T: 5, 10, 15, 20, and 30	Different roasted temperatures and times	138 °C roasted 6 min	97.23% and 92.34%	[89]
8	Robusta	T: 220–260	Supercritical CO_2_ extraction	100 °C, 200 Pa, 9.5% ethanol solution for 1035 min	79%	[91]

## Data Availability

Available on request and with regulations.

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
