# Peer review of "Production and Inhibition of Acrylamide during Coffee Processing: A Literature Review"

_molecules, 2023, doi:10.3390/molecules28083476_

Round 1
Reviewer 1 Report
The formation and presence of acrylamide in coffee is a public health concern. For that reason it is important to address this aspect, evaluate and expand knowledge about this food matrix in order to understand the formation mechanisms and seek mitigation strategies to ultimately reduce the risk associated with exposure to the contaminant. Authors have carried out a bibliographical review on this subject including different aspects, however, some of them should be somewhat expanded in order to improve the complete review. Some considerations are mentioned below:
- English style should be checked throughout the manuscript.
- Acrylamide formation from dicarbonyl compounds should be considered.
- Line 60: I suggest replacing "Figure 1a-c" by "Figure 1".
- Section 2.1. Sugar caramelization should also be considered for the contribution to the formation of HMF
- Mutagenicidad and genotoxicity of acrylamide and its active metabolite glycidamide should also be considered. I recommend the authors to mention the latest EFSA report that addresses these issues.
https://www.efsa.europa.eu/en/efsajournal/pub/7293
- European Regulation about acrylamide in foods and benchmark levels stablished for coffee should be mentioned.
- Influence of agronomical factors to the precursor composition in green coffee should be mentioned.
- Is the reduction in acrylamide levels due to the presence of SH- groups related to the formation of Michael addition compounds? please, reconsider this aspect
- The reduction of acrylamide during coffee storage can also be produced by binding to other compounds such as melanoidins. I suggest including this aspect in the review
- I recommend mentioning decaffeinated coffee and its acrylamide content
- Acrylamide exposure through the consumption of coffee could also be mentioned
- Other mechanisms of acrylamide reduction in coffee such as the application of enzymes other than asparaginase, supercritial CO2 or steam roasting should be indicated.
Reviewer 2 Report
In the present manuscript, the mechanisms which lead to the production of acrylamide and other undesidered compounds during the coffee processing are reviewed, together with the possible strategies to decrease their amount. The toxicity of AA is adequately considered and discussed.
I think that a language check and errors correction is necessary. Some other comments are as follows:
Introduction
- Please check the correspondance between the mentioned references and the text (ref 16 does not mention asparaginase, as an example)
- Line 38: please omit “excessive”
- Line 43: please omit “unnecessary”
Section 2.:
Line 58: why “consequently”?
Line 78: please delete “mellard”
2.3, lines 142-145: please clarify, maybe it is difficult to use HPLC or spectrophotometry to detect AA?
Lines 150-156: have those new detection methods been validated for acrylamide detection? Are they able to establish AA concentrations?
Lines 159-163: chromatographic conditions are not adequately described: how was the separation performed?
Section 3: in my opinion, this section is the strength of the manuscript.
I appreciate Figure 3, is very clear and informative.
Round 2
Reviewer 1 Report
Authors have considered all the comments suggested by the reviewer. Manuscript has been improved and it will be suitable to be published in the present form